# Consequences of Contracting COVID-19 or Taking the COVID-19 Vaccine for Individuals with a History of Lyme Disease

**DOI:** 10.3390/antibiotics12030493

**Published:** 2023-03-01

**Authors:** Daniel J. Cameron, Sean R. McWhinney

**Affiliations:** 1Northern Westchester Hospital, Mt. Kisco, New York, NY 10549, USA; 2Department of Psychiatry, Dalhousie University, Halifax, NS B3H 2E2, Canada

**Keywords:** COVID-19, Lyme disease, long COVID, COVID-19 vaccine

## Abstract

Individuals with Lyme disease can be very symptomatic. This survey compares the burden of illness for individuals with a history of Lyme disease (HLD) with individuals with a HLD who have either contracted COVID-19 or who have taken the COVID-19 vaccine. The findings describe the relative symptom burden among these three groups using a cross-sectional descriptive survey investigating the burden of Lyme disease in a pandemic. The survey includes the General Symptom Questionnaire-30 (GSQ-30), a brief self-report scale designed to assess the symptom burden in Lyme disease (LD). The results of this survey show that the overall burden of illness among individuals with HLD is not significantly different after contracting COVID-19 or after COVID-19 vaccination. A new survey will be needed to better understand why one in five individuals with a HLD reported long COVID after contracting COVID-19. These results should help clinicians and their patients to discuss the consequences of contracting a COVID-19 infection or being vaccinated against COVID-19.

## 1. Introduction

Lyme disease (LD) can lead to Lyme encephalopathy [1,2], Lyme neuropathy [3], neuropsychiatric Lyme disease [4], pediatric acute-onset neuropsychiatric syndrome (PANS) [5], Lyme carditis [6], Postural Orthostatic Tachycardia Syndrome (POTS) [7], and Post-Treatment Lyme Disease Syndrome (PTLDS) [8]. Chronic neurologic Lyme disease cases were ill for up to 14 years [1]. Lyme encephalopathy patients enrolling in an NIH-sponsored trial were ill for an average of nine years [2]. Patients with persistent symptoms and a history of chronic Lyme disease enrolled in another NIH trial were ill for an average of 4.7 years [9].

Their symptoms can be severe. The quality of life on the SF-36 scale for individuals with persistent symptoms enrolling in an NIH trial was lower than for individuals with diabetes and heart disease. Individuals with PTLDS were worse than individuals with traumatic brain injury (TBI), depression, and Erythema Migrans (EM) using a Multi-System Symptom Burden (MSSB) Scale [10].

Their function can be poor. Of 27 individuals with neurologic Lyme disease, three quit their jobs, three reduced their work load to part time, and two retired early [1]. Twenty-five adolescents with Lyme disease had deficits in cognition and worse attendance, grades, and subjective reports of memory problems that far exceeded those of the controls [11]. Individuals with PTLDS experience fatigue, pain, and cognitive difficulties and poor function that last more than six months post-treatment [8]. Some professionals have attributed the symptoms and loss of function in PTLDS to other illnesses, such as fibromyalgia, chronic fatigue, and mood disturbances.

Individuals with a HLD remain ill despite antibiotic treatment. Of 15 individuals with Lyme encephalopathy, five said that they were now normal, seven were greatly improved, and three were somewhat improved [12]. Individuals with Lyme encephalopathy and individuals with persistent symptoms with a history of Lyme disease enrolling in two NIH-sponsored trials failed antibiotic treatment [2,9]. Individuals with Post Lyme Disease enrolling in a third NIH-sponsored trial were successfully treated for fatigue [13]. Individuals with PTLDS can remain ill despite antibiotic treatment [8].

COVID-19 can lead to symptoms in these same domains. Aiyegbusi and colleagues described cognitive impairment, memory loss, anxiety, and sleep disorders [14]. Furthermore, Crook and colleagues described fatigue, dyspnea, cardiac abnormalities, cognitive impairment, sleep disturbances, symptoms of post-traumatic stress disorder, muscle pain, concentration problems, and headache [15].

COVID-19 can lead to functional impairments. Up to one out of three individuals with a history of COVID-19 in a literature search had a decreased functional capacity on at least one measure. The authors were not able to determine the factors leading to the reduced functional capacity [16]. Their literature search could not determine the duration of the functional impairment.

Receiving a COVID-19 vaccination can lead to some of these symptoms in these same domains. The most common side effects in the clinical trial data were injection site pain, headache, and fever, with most side effects lasting fewer than seven days [17]. It is unclear whether a COVID-19 vaccine can lead to a loss of function.

It has not been documented whether the effects of COVID-19 might uniquely impact those with a history of Lyme disease (HLD). This survey examines the symptom burden for individuals with HLD relative to those with HLD who have either contracted COVID-19 or who have taken the COVID-19 vaccine. We examined four individual symptom domains (pain/fatigue, neuropsychiatric, neurological, and viral-like symptoms) and the overall symptom burden to identify differences between these groups of individuals.

## 2. Results

### 2.1. Sample

Sample characteristics are shown in Table 1. Briefly, this included 889 (15.9% male) individuals (289 with a history of LD, 174 who had contracted COVID-19, and 426 who had been vaccinated). The percentage of HLD vaccinated included Moderna (34.7%), Pfizer (54.9%), Johnson & Johnson (8.2%), and other vaccines (2.1%). Most of the respondents (89.8%) received two vaccines. The boosters were not available for the respondents at the time of the survey. The mean age was 50.5 years (SD = 13.9). The vast majority of individuals with a HLD were ill for more than 4 weeks (98.3%). Slightly over one third (37.2%) had been diagnosed with PTLDS. Importantly, the three groups significantly differed in their rates of fibromyalgia (χ^2^ = 8.23, *p* = 0.017). Moreover, significantly worse symptoms (higher total GSQ score) were found in those with fibromyalgia (F(1168) = 9.69, *p* = 0.002) and chronic fatigue (F(1168) = 16.01, *p* < 0.001). We, therefore, controlled for the presence of either fibromyalgia or chronic fatigue, as well as participant age and sex, in all analyses of group differences.

### 2.2. HLD Who Contracted COVID-19

The MSSB of HLD without contracting COVID-19 and without vaccination against COVID-19 was 46.34 (SD = 27.58).

Overall, the MSSB of HLD who contracted COVID-19 was not significantly greater than for HLD who had not contracted COVID-19 (F(1456) = 0.04, *p* = 0.843). Moreover, the two groups did not significantly differ in any of the four domains of the GSQ (see Table 2, Figure 1). While there was variability between domains, patterns of variability were similar for the two groups.

Approximately one in five (17%) individuals with a HLD who contracted COVID-19 reported long COVID on a post-hoc analysis. The incidence of long COVID was not significantly different between Moderna (38.5%), Pfizer (38.9%), Johnson & Johnson (45.7%), and other vaccines (44.4%), respectively (χ^2^ = 0.75, *p* = 0.860). The MSSB of HLD who suffered from long COVID was significantly worse than for those who did not (F(1,3) = 6.35, *p* = 0.013). This difference was driven by worse neurological symptoms (Table 2, Figure 1).

### 2.3. HLD Who Were Vaccinated against COVID-19

Overall, the MSSB of HLD who were vaccinated against COVID-19 was not significantly greater than for HLD who had not been vaccinated (F(1705) = 2.89, *p* = 0.089). However, when considering specific domains, there was a significant interaction between group and domain (F(6,2658) = 3.06, *p* = 0.005). Specifically, post-hoc testing showed that those who had been vaccinated varied little in pain and fatigue, neuropsychiatric, and viral-like symptoms but did show significantly fewer neurological symptoms than those with HLD (Table 2).

## 3. Methods

### 3.1. Sampling

These findings summarize the results of a cross-sectional descriptive survey investigating the burden of Lyme disease in a pandemic. Participants were recruited through a snowball sampling strategy. This survey was conducted in accordance with the Declaration of Helsinki and approved by the Western Institutional Review Board. Participants were eligible for inclusion if they had HLD, whether or not they had contracted COVID-19 or had taken the COVID-19 vaccine. Respondents who had both contracted COVID-19 and taken the vaccine were excluded to avoid the confounding of these two effects. The result was three groups: one with HLD, a second who had contracted COVID-19 but had not been vaccinated, and a third who had been vaccinated but had not contracted COVID-19.

### 3.2. Surveys

Participants were asked to describe their demographic and social factors (age, sex, marital state, and working status). Moreover, they were asked to describe their Lyme disease-related factors, including a history of a tick bite, late manifestations of Lyme disease, a history of Lyme disease and associated tick-borne illness, laboratory tests for Lyme disease and associated tick-borne infections, and the number and type of chronic diseases or illnesses, injuries, hospitalizations, surgeries, allergies, and health habits.

We also assessed participants′ history of COVID-19, the length and severity of ongoing effects, and their outcomes. COVID-19 vaccine-related factors included a history of a COVID-19 vaccine or their hesitancy if any. Participants were allowed to expand on their experience with COVID-19, or with the COVID-19 vaccine, to supplement quantitative data with a qualitative explanation.

We used the General Symptom Questionnaire-30 (GSQ-30) [10] to contrast the symptom burden between groups. The GSQ-30 is a brief self-report scale designed to assess the MSSB in Lyme disease. The MSSB is a validated measure of pain/fatigue, neuropsychiatric, neurological, and viral-like symptoms [10]. The MSSB was worse for individuals with PTLDS (M = 42.38, SD = 22.14) than for individuals with traumatic brain injury (TBI) (M = 32.82, SD = 26.79), depression (M = 42.28, SD = 21.05), and Erythema Migrans (EM) (M = 24.15, SD = 20.11) [10].

The GSQ-30 asks “How much have you been bothered by any of the following?” for each of 30 items, which in tandem assess four domains, with each item rated on a five-point Likert scale. Responses include “Not at all”, “A little bit”, “Somewhat”, “Quite a bit”, and “Very much” (scored 0–4, summed in each domain and overall for a total score of up to 120). Each domain is relevant to specific symptoms in LD (pain/fatigue, neuropsychiatric, neurologic, and viral-like symptoms) [10].

The survey included a hypothesis-generating or exploratory question, “Have you had any of the following chronic conditions from your COVID-19 illness?” The responses consisted of pneumonia, renal failure, clotting disorder, shortness of breath, loss of smell, loss of taste, long hauler (i.e., long COVID), positional Orthostatic Tachycardia Syndrome (POTS), other, don′t know, not sure, prefer not to answer.

### 3.3. Statistical Analysis

Statistical modeling was completed using R version 4.1.1, with linear mixed models using the package *lme4* [18]. GSQ-30 scores per domain were tested for associations with groups (HLD, COVID, or Vaccine), GSQ domain (pain/fatigue, neuropsychiatric, neurological, or viral-like), and an interaction between the two, with covariates controlling for age and the presence of any comorbidities that significantly differed between groups. Among those with reported COVID-19 infections, we tested this same model, but with the effect of the presence of long COVID instead of group differences. We included a random grouping factor for each participant to control for repeated measures in all models. Post-hoc testing was completed using the R package *emmeans* using Tukey′s HSD [19]. Multicollinearity was evaluated by calculating the Variance Inflation Factor (VIF) and was found to be negligible. Model residual normality was ensured using QQ plots.

## 4. Discussion

This survey confirms the symptom burden and duration of illness of individuals with a HLD. The symptom burden of HLD without contracting COVID-19 or vaccination against COVID-19 was worse than in previous reports of individuals with an EM rash, TBI, and PTLDS. The duration of illness of 14.5 years (SD = 13.67) was longer than the 4.7 and 9 years, respectively, reported in two NIH-sponsored trials [2,9].

The symptom burden of HLD was not affected by individuals who contracted COVID-19 or were vaccinated against COVID-19. However, on post-hoc analysis, those who were vaccinated against COVID-19 had significantly lower neurological symptoms. The neurologic symptoms consisted of hot or cold sensations in extremities, skin or muscle twitching, numbness or tingling, bladder discomfort or change in urination, balance problems or a sense of room spinning, change in visual clarity or trouble focusing, shortness of breath, irregular or rapid heartbeat, shooting, stabbing, or burning pains, and discomfort with normal light or sound. The causes for the lower neurologic symptoms in HLD who were vaccinated and the time course of their resolution should be examined in future studies to determine the extent of their impact on quality of life.

Nearly one in five HLD individuals who contracted COVID-19 reported long COVID. This finding is exploratory, as the issue is new [20]. The diagnostic criteria, etiology, and natural course of long COVID have not been established at the time of publication. Subramanian and colleagues reported in 2022 that 5.4% of individuals with a history of SARS and 4.3% without a history of SARS described at least one of 33 symptoms defined by the World Health Organization’s clinical case definition 12 weeks after the onset of illness [20].

On post-hoc analysis, those who contracted long COVID varied little in pain and fatigue, neuropsychiatric, and viral-like symptoms but did show significantly higher neurological symptoms. The specific mechanisms of long COVID are still unclear and require further investigation.

One in three participants with HLD described a history of PTLDS. This is higher than the 10 to 20% risk of PTLDS described in previous trials in patients optimally treated for early Lyme disease [21]. The risk of PTLDS may be higher in actual practice to explain the high symptom burden in survey participants who were not diagnosed with PTLDS.

Comorbidities to consider for long COVID have included COPD, benign prostatic hyperplasia, fibromyalgia, anxiety, erectile dysfunction, depression, migraine, multiple sclerosis, celiac disease, and learning disabilities [20]. This survey suggests the need to examine HLD as a comorbidity for long COVID.

The strengths of this survey include the large sample size and the use of a validated measure of the symptom burden of Lyme disease. The primary limitation of this survey is that it is unclear why one in five HLDs who contracted COVID-19 reported long COVID, as the survey was not designed to evaluate the possible causes of long COVID. In other words, a cross-sectional approach does not provide evidence for causality as the data represent a snapshot in time. A longitudinal design would be helpful to determine the time course and potential causality of these relationships. Other limitations include, first, that the MSSB measure was designed for HLD before the pandemic and not for HLD who had contracted COVID-19 or taken the COVID-19 vaccine. Second, the exact mechanism behind the MSSB has to be identified to determine if the ongoing symptoms of long COVID are related to COVID-19 or a persistent tick-borne infection. Third, the addition of at least one more non-HLD COVID-19 population would be helpful to understand the consequences of contracting COVID-19 or taking the COVID-19 vaccine. Fourth, the timing between either contracting COVID-19 or being vaccinated against COVID-19 and completing the survey was not known. Lastly, the survey may have been subject to volunteer bias, where participants who actively decided to participate in the research may have differed from the general population of individuals with HLD.

## 5. Conclusions

The results of this survey have shown that individuals with HLD can be significantly ill in four individual symptom domains (pain/fatigue, neuropsychiatric, neurological, and viral-like symptoms). Individuals with HLD were not significantly different in these four domains of the overall burden of illness after contracting COVID-19 or after COVID-19 vaccination. A new survey will be needed to better understand why one in five individuals with a HLD reported long COVID after contracting COVID-19. These results should help clinicians and their patients to discuss the consequences of contracting a COVID-19 infection or being vaccinated against COVID-19.

## Figures and Tables

**Figure 1 antibiotics-12-00493-f001:**
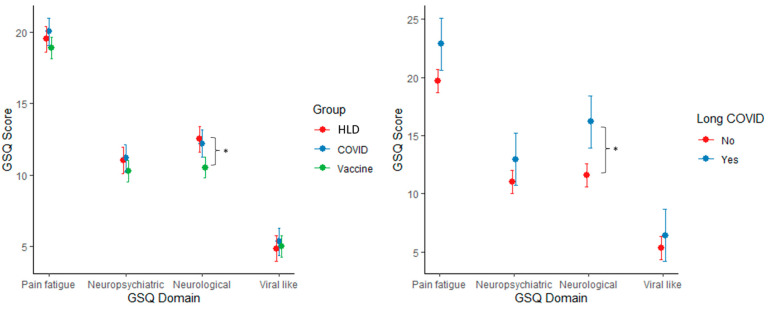
GSQ domain scores in each group, controlling for age, sex, and fibromyalgia or chronic fatigue. Significant differences are shown using asterisks (*, *p* < 0.05).

**Table 1 antibiotics-12-00493-t001:** Sample characteristics overall and by group.

	Overall	HLD	HLD with COVID-19	HLD with COVID-19 Vaccine
Sample size (n)	889	289	174	426
Age—mean (SD)	50.50 (13.92)	50.33 (13.75)	45.26 (13.36)	52.76 (13.70)
Sex—M (%)	140 (15.9)	38 (13.2)	34 (19.7)	68 (16.1)
Vaccine willingness—Yes (%)	743 (83.6)	210 (72.7)	107 (61.5)	426 (100.0)
GSQ				
Total—mean (SD)	43.88 (27.43)	46.34 (27.58)	46.99 (28.32)	40.95 (26.72)
Pain/fatigue—mean (SD)	19.28 (11.02)	19.78 (11.03)	20.26 (10.95)	18.55 (11.02)
Neuropsychiatric—mean (SD)	10.61 (6.61)	11.26 (6.51)	11.15 (6.66)	9.95 (6.60)
Neurological—mean (SD)	11.48 (9.48)	12.77 (9.74)	12.46 (10.00)	10.20 (8.92)
Viral-like—mean (SD)	5.02 (4.37)	5.11 (4.50)	5.69 (4.67)	4.69 (4.12)
Comorbidities				
Any—Yes (%)	600 (67.5)	195 (67.5)	123 (70.7)	282 (66.2)
Fibromyalgia—Yes (%)	237 (26.7)	91 (31.5)	51 (29.3)	95 (22.3)
Chronic fatigue—Yes (%)	406 (45.7)	140 (48.4)	88 (50.6)	178 (41.8)
Fibromyalgia or chronic fatigue—Yes (%)	441 (49.6)	155 (53.6)	95 (54.6)	191 (44.8)
PTLDS—Yes (%)	331 (37.2)	96 (33.2)	67 (38.5)	168 (39.4)
Mood disorder—Yes (%)	166 (18.7)	55 (19.0)	36 (20.7)	75 (17.6)
Attention-deficit/hyperactivity disorder (ADHD)—Yes (%)	114 (12.8)	37 (12.8)	25 (14.4)	52 (12.2)
Lupus—Yes (%)	15 (1.7)	3 (1.0)	5 (2.9)	7 (1.6)
Rheumatoid arthritis—Yes (%)	60 (6.7)	17 (5.9)	15 (8.6)	28 (6.6)
Multiple sclerosis—Yes (%)	15 (1.7)	6 (2.1)	2 (1.1)	7 (1.6)
Mastocytosis—Yes (%)	8 (0.9)	5 (1.7)	0 (0.0)	3 (0.7)

**Table 2 antibiotics-12-00493-t002:** Comparison of GSQ domain scores between groups, controlling for age, sex, and fibromyalgia or chronic fatigue. Tests including long COVID (^a^) are only among those with reported COVID-19 infections. Significant differences are shown using asterisks (*, *p* < 0.05).

Domain	Group	Estimate	SE	DF	t	*p*	Significant
Pain/fatigue	HLD vs. COVID-19	−0.32	0.76	1637	−0.42	1.000	
	HLD vs. Vaccine	0.76	0.60	1641	1.26	0.984	
	COVID-19 vs. Vaccine	1.08	0.72	1615	1.50	0.941	
	Long COVID vs. COVID-19 ^a^	−3.36	1.33	324	−2.52	0.190	
Neuropsychiatric	HLD vs. COVID-19	0.27	0.76	1637	0.35	1.000	
	HLD vs. Vaccine	0.84	0.60	1641	1.38	0.967	
	COVID-19 vs. Vaccine	0.57	0.72	1615	0.78	1.000	
	Long COVID vs. COVID-19 ^a^	−2.30	1.33	324	−1.73	0.668	
Neurological	HLD vs. COVID-19	0.47	0.76	1637	0.61	1.000	
	HLD vs. Vaccine	2.09	0.60	1641	3.46	0.028	*
	COVID-19 vs. Vaccine	1.62	0.72	1615	2.24	0.518	
	Long COVID vs. COVID-19 ^a^	−4.75	1.33	324	−3.57	0.010	*
Viral-like	HLD vs. COVID-19	−0.42	0.76	1637	−0.55	1.000	
	HLD vs. Vaccine	−0.06	0.60	1641	−0.10	1.000	
	COVID-19 vs. Vaccine	0.36	0.72	1615	0.50	1.000	
	Long COVID vs. COVID-19 ^a^	−1.09	1.33	324	−0.82	0.992	

## Data Availability

Anonymized data will be provided upon reasonable request.

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
