# Peer review of "Consequences of Contracting COVID-19 or Taking the COVID-19 Vaccine for Individuals with a History of Lyme Disease"

_antibiotics, 2023, doi:10.3390/antibiotics12030493_

Round 1

Reviewer 1 Report (New Reviewer)

In this study, the authors examine the consequences for individuals with a history of Lyme disease (HLD), as well as those who have either contracted COVID-19 or who have taken the COVID-19 vaccine. The findings describe the relative symptom burden among these three groups in COVID-19 pandemic. This is an useful survay, suggesting the potential benefit for vaccination in Lyme population, and could assist physicians in evaluating HLDs who have contracted COVID-19 or been vaccinated against COVID-19.

Author Response

I have tightened up the paper based on reviews. I focused on the overall symptoms burden rather than the post-hoc findings.  The left post-hoc observations as findings supportive of future study.

"The results of this survey have shown that the overall burden of illness among individuals with HLD was not significantly different after contracting COVID-19 or after COVID-19 vaccination. Nearly, one in five HLDs who contracted COVID-19 reported long COVID.  These results can also help clinicians and their patients understand the consequence of contracting a COVID-19 infection or being vaccinated against COVID-19."

"

Reviewer 2 Report (New Reviewer)

This manuscript is so seriously flawed that in my view it should not be further considered for being published. Introduction contains only a few long lists of symptoms without further explanation. PTLDS and MSSB are at least questionable for a number of researchers but the authors do not discuss this issue. Statistics regarding side effects of COVID-19 vaccines are presented in a very subjective way without comparing their effectiveness to other vaccines and without differentiating between mild and severe adverse events.

Methods do not define the exact reliable criteria of participants' categorization. How do the authors know if HLD patients did not develop oligosymptomatic course of COVID-19? It is a very strong methodology bias and in my view it is impossible to be excluded through a survey. Furthermore, fibromyalgia appears here out of sudden. There is not a single mention of fibromyalgia in the Introduction section.

The Results are very messed and the statistics seem to be mischiefed. The majority of the presented results were not statistically significant.

Discussion contains a website link sending directly to the author's website. I am not sure if this is acceptable in MDPI policy?

Conclusions are not supported by the content. "These results could assist physicians in evaluating HLDs who have contracted COVID-19 or been vaccinated against COVID-19 by identifying specific symptoms domains that are likely to be affected". Such recommendation requires further elaboration before being published elsewhere.

Author Response

This manuscript is so seriously flawed that in my view it should not be further considered for being published.

These comments offered new insights into the paper.  I had been caught in the post-hoc analysis rather than the primary issues of overall symptoms severity. I had to make more changes that I anticipated.  I appreciate the opportunity to address the latest comments. 

Introduction contains only a few long lists of symptoms without further explanation. PTLDS and MSSB are at least questionable for a number of researchers but the authors do not discuss this issue.

The paper better defines PTLDS while acknowledging the controversy.  The rationale for a validated measure of symptoms in this population was included in the introduction.

Statistics regarding side effects of COVID-19 vaccines are presented in a very subjective way without comparing their effectiveness to other vaccines and without differentiating between mild and severe adverse events.

We pulled the side effect topic to allow us to expand on the topic in another paper.

Methods do not define the exact reliable criteria of participants' categorization. How do the authors know if HLD patients did not develop oligosymptomatic course of COVID-19? It is a very strong methodology bias and in my view it is impossible to be excluded through a survey.

This issue is addressed as follows in the discussion. “Second, the exact mechanism behind the MSSB has to be identified to determine if the ongoing symptoms of long COVID were related to COVID-19 or persistent tick-borne infection.”

There is not a single mention of fibromyalgia in the Introduction section.

Fibromyalgia, chronic fatigue and mood issues were included in the introduction.

The Results are very messed and the statistics seem to be mischiefed. The majority of the presented results were not statistically significant.

The reviewers comments afforded us to better understand the Survey results. I am afraid we were lost in the weeds during the post-hoc analysis. 

Discussion contains a website link sending directly to the author's website. I am not sure if this is acceptable in MDPI policy?

The link to outside data was no longer necessary.  The link to the author’s website was removed.

Conclusions are not supported by the content. "These results could assist physicians in evaluating HLDs who have contracted COVID-19 or been vaccinated against COVID-19 by identifying specific symptoms domains that are likely to be affected". Such recommendation requires further elaboration before being published elsewhere.

The conclusions and abstract were limited to comments that were supported by the evidence.  “The results of this survey have shown that the overall burden of illness among individuals with HLD was not significantly different after contracting COVID-19 or after COVID-19 vaccination. Nearly, one in five HLDs who contracted COVID-19 reported long COVID.  These results can also help clinicians and their patients understand the consequence of contracting a COVID-19 infection or being vaccinated against COVID-19.”

Reviewer 3 Report (Previous Reviewer 3)

Should be specified 1) in the introduction the specifity of immune response of LD infection, 2) how were vaccinated the investigated individuals (fully vaccinated, boosted) and side effects in these separate groups. 

Author Response

Should be specified 1) in the introduction the specificity of immune response of LD infection,

The Survey was not designed to examine the specificity of the immune response.  The Survey should encourage clinical research.

2) how were vaccinated the investigated individuals (fully vaccinated, boosted) and side effects in these separate groups.

We pulled the side effect topic to allow us to expand on the topic in another paper.

Reviewer 4 Report (New Reviewer)

Sorry, but I suggest to read the paper once again and please add more information on Lyme Disease patients. Please consider also the conclusions - patients with neurological and other, included at work, symptoms are common in out-patients clinic and to help them is difficult. 

Author Response

Dear reviewer:

We appreciate your review and the following thoughtful comment.

Sorry, but I suggest to read the paper once again and please add more information on Lyme Disease patients. Please consider also the conclusions - patients with neurological and other, included at work, symptoms are common in out-patients clinic and to help them is difficult. 

We added the following information on Lyme disease to allow the reader to understand the morbidity associated with Lyme disease, particularly the duration and severity of Lyme disease, the impact on function, and the problem with patients failing Treatment for Lyme disease.

Lyme disease (LD) can lead to Lyme encephalopathy,1,2 Lyme neuropathy,3 Neuropsychiatric Lyme disease,4 Pediatric acute-onset neuropsychiatric syndrome (PANS),5 Lyme carditis,6 Postural Orthostatic Tachycardia Syndrome (POTS)7, and Post Treatment Lyme disease Syndrome (PTLDS).8  Chronic neurologic Lyme disease cases were ill up to 14 years.1 Lyme encephalopathy patients enrolling in a NIH sponsored  trial were ill an average of nine years.2 Patients with persistent symptoms and a history of Lyme disease chronic enrolled in another NIH trial were ill an average of 4.7 years.9

Their symptoms can be severe.  The quality of life of a SF-36 scale for individuals with patients with persistent symptoms enrolling in a NIH trial were sicker than for individuals with diabetes and heart disease.  Individuals with PTLDS were worse than individuals with traumatic brain injury (TBI) depression and an Erythema Migrans (EM) using a Multi-System Symptom Burden (MSSB) Scale.10

Their function can be poor. Of 27 individuals with neurologic Lyme disease, three quit their jobs, three reduced their work load to part times, and two retired early.1 Twenty-five adolescents with Lyme disease had deficits in cognition, worse attendance, grades, and subjective reports of memory problems that far exceeded the controls.11 Individuals with PTLDS experience fatigue, pain, and cognitive difficulties and poor function that last more than six months post-treatment.8 Some professionals have attributed the symptoms and loss of function in PTLDS to other illnesses such as fibromyalgia, chronic fatigue, a mood disturbance.

Individuals with a HLD remain ill despite antibiotic treatment. Of 15 individuals with Lyme encephalopathy 5 said that they were now normal, 7 were greatly improved, and 3 were somewhat improved.12 Individuals with Lyme encephalopathy and individuals with persistent symptoms with a history of Lyme disease enrolling in two NIH sponsored trials failed antibiotic treatment.2,9 Individuals with Post Lyme Disease enrolling in a third NIH sponsored trial were successfully treated for fatigue.13 Individuals with PTLDS can remain ill despite antibiotic treatment.8

We also modified the last two sentences of the conclusion  and abstract to clarify the relevance of the findings for doctors in out patients clinics.

This manuscript is a resubmission of an earlier submission. The following is a list of the peer review reports and author responses from that submission.

Round 1

Reviewer 1 Report

This survey study titled “Consequences of contracting COVID-19 or taking the COVID-19 vaccine for individuals with a history of Lyme disease” by  Cameron et al., aimed at identifying  specific symptom domains that may be affected be either contracting COVID-19 or by taking the COVID-19 vaccine in those with HLD. The study made use of General Symptom Questionnaire-30 (GSQ-30), a brief self-report scale that was designed to assess the symptom burden in Lyme disease (LD). They identified that the neurological symptoms were significantly worse both in those who had not been vaccinated relative to those who had, as well as in those with long-COVID relative to those without. The authors suggest that this study would help with shared decision making for HLD who have not contracted COVID-19 or taken the vaccine.

Comments:

The strength of the study is that it has a strong sample size of 889 individuals with important variables considered. The study also included the main symptoms of individuals with HLD and its relation to COVID. They also make use of a validated measure of the MSSB of Lyme disease. They study is well designed and the result from this is an important step towards determining the benefits of HLD individuals to get vaccinated COVID.

Author Response

We appreciate your thoughtful review.  We also hope this information will help doctors and their patients who need to make decisions regarding the COVID-19 vaccine.  

Reviewer 2 Report

The Manuscript written by Daniel J. Cameron and Sean R. McWhinney touches on the interesting topic of comparing symptoms after suffering Lyme disease with symptoms associated with Covid-19 and after Covid-19 vaccination. But this study is not accurate enough, because there is no group of people who recovered from Covid-19, without HLD.

There is no information about when the participants had Lyme disease, what is the time interval between Lyme disease and COVID or vaccination.

What does COVID vaccinated mean? Is it two doses of the vaccine, or 2 doses and a booster or 2 boostres?

The introduction provides data on numerous side effects after vaccination. And vaccines are not uniform in the frequency of these effects. It is not clear what vaccines the participants were vaccinated with.

Also, these side effects disappear within 7 days (Introduction), but in the results you highlight groups (32.6% and 4%) in which these effects lasted more than 3 days. If they pass in 7 days, why single out groups in which they lasted more than 3 days. If side effects lasted more than 7 days, then it would be necessary to single them out. And maybe it is related to the type of vaccine and these 32.6% and 4% of the participant received a specific vaccine?

In the Introduction, the authors describe that Lyme disease can lead to acute and chronic illness and sometimes leads to Post Treatment Lyme disease Syndrome. Consequently, another part of patients, after receiving appropriate treatment, recover completely. But in Table 1 we see that only 37% of the people in this study have a history of the Post Treatment Lyme disease Syndrome and what causes 4 groups of symptoms in the other 63% of participants is not clear.

Results

In Table 1: what is ADHD? I didn’t find the definition of the acronym.

In Table 1: there is no COVID-only group without Lyme disease in history.

In 3.2. “…COVID-19 was 46.34 (SD=27.58), which was worse than patients with a history of an Erythema Migrans (EM) (M=24.15, SD=20.11), PTLDS (M=42.38, SD=22.14), traumatic brain injury (TBI) (M=32.82, SD=26.79) and depression (M=42.28, SD=21.05).” Where did these numbers come from? Here is a link to reference 1. But how can you compare your data with data from another study? These comparisons should be moved to the Discussion.

In Table 2: “….long COVID…” What does it mean long COVID? How many days or months do long COVID last? For some, a week is a lot, but for others, a month is not enough.

In 3.3. “One HLD who reported a severe side effect lasting more than three days improved after being administered the COVID-19 vaccine.” Was this patient revaccinated or was he still vaccinated?

Author Response

We appreciate your thoughtful comments and guidance on our manuscript. The manuscript is now clearer in response to your review.

The Manuscript written by Daniel J. Cameron and Sean R. McWhinney touches on the interesting topic of comparing symptoms after suffering Lyme disease with symptoms associated with Covid-19 and after Covid-19 vaccination. But this study is not accurate enough, because there is no group of people who recovered from Covid-19, without HLD.

The Survey was designed to compare individuals a HLD with and without COVID-19, and not with the general population. This Survey would support a future Survey comparing individuals with a HLD and COVID-19 with the general population.

There is no information about when the participants had Lyme disease, what is the time interval between Lyme disease and COVID or vaccination.

The following information was added. “The participants with HLD in this Survey were ill an average of 14.5 years (SD=13.67).  Individuals with a HLD were ill an average of 4.7 and 9 years respectively in two NIH sponsored trials.[3, 14]”. Unfortunately, while we cannot know the interval between contracting Lyme disease and COVID-19, we are not aware of any evidence to suggest that this factor would alter the severity of side effects.

What does COVID vaccinated mean? Is it two doses of the vaccine, or 2 doses and a booster or 2 booster?

Thank you for highlighting this oversight. The following information was added. “The percent of HLD vaccinated was included Moderna (34.7%), Pfizer (54.9%), Johnson & Johnson (8.2%) and other vaccines (2.1%%) respectively. Most of the respondents (89.8%) received two vaccines. The boosters were not available for the respondents at the time of the Survey.”

The introduction provides data on numerous side effects after vaccination. And vaccines are not uniform in the frequency of these effects. It is not clear what vaccines the participants were vaccinated with.

The following information was added. “The incidence of side effects was not significantly different between Moderna (84.5%), Pfizer (79.5%), Johnson & Johnson (88.6%) and other vaccines (77.8%) respectively (χ2=2.73, p=0.435).”

Also, these side effects disappear within 7 days (Introduction), but in the results you highlight groups (32.6% and 4%) in which these effects lasted more than 3 days. If they pass in 7 days, why single out groups in which they lasted more than 3 days. If side effects lasted more than 7 days, then it would be necessary to single them out. And maybe it is related to the type of vaccine and these 32.6% and 4% of the participant received a specific vaccine?

The frequency of side effects for the COVID-19 vaccine for individuals with HLD may be less if the Survey had assessed side effects at 7 days. We therefore selected a shorter time frame to compare between vaccine types, as you’ve suggested, and have demonstrated there to be no significant difference between vaccine types. 

The following information was added. The incidence of side effects was not significantly different between Moderna (84.5%), Pfizer (79.5%), Johnson & Johnson (88.6%) and other vaccines (77.8%) respectively (χ2=2.73, p=0.435).

In the Introduction, the authors describe that Lyme disease can lead to acute and chronic illness and sometimes leads to Post Treatment Lyme disease Syndrome. Consequently, another part of patients, after receiving appropriate treatment, recover completely. But in Table 1 we see that only 37% of the people in this study have a history of the Post Treatment Lyme disease Syndrome and what causes 4 groups of symptoms in the other 63% of participants is not clear.

Thank you for your comments, we have attempted to clarify that even in patients who did not develop PTLDS, some degree of side effects and/or symptoms of COVID-19 are still expected. The following information was added. “The vast majority of individuals with a HLD were ill for more than 4 weeks (98.3%). Just over one third (37.2%) had been diagnosed with PTLDS.”

“One in three participants with HLD described a history of PTLDS. That is higher than the 10 to 20% risk of PTLDS described in previous trials in patients optimally treated for early Lyme disease.[15] The risk of PTLDS may be higher in actual practice to explain the high MSSB in Survey participants who not diagnosed with PTLDS.”

The Survey would support research to understand the incidence of PTLDS in individuals who are not optimally treated for Lyme disease

Results

In Table 1: what is ADHD? I didn’t find the definition of the acronym.

Added “Attention-deficit/hyperactivity disorder (ADHD)

In Table 1: there is no COVID-only group without Lyme disease in history.

As noted previously, the Survey was designed to compare individuals with a HLD with and without COVID-19, and not with the general population. This Survey would support a future Survey comparing individuals with a HLD and COVID-19 with the general population.

In 3.2. “…COVID-19 was 46.34 (SD=27.58), which was worse than patients with a history of an Erythema Migrans (EM) (M=24.15, SD=20.11), PTLDS (M=42.38, SD=22.14), traumatic brain injury (TBI) (M=32.82, SD=26.79) and depression (M=42.28, SD=21.05).” Where did these numbers come from? Here is a link to reference 1. But how can you compare your data with data from another study? These comparisons should be moved to the

The MSSB was moved to the introduction as background information:  “The MSSB was worse for PTLDS (M=42.38, SD=22.14) than for traumatic brain injury (TBI) (M=32.82, SD=26.79), depression (M=42.28, SD=21.05), and an Erythema Migrans (EM) (M=24.15, SD=20.11) using a validated General Symptom Questionnaire-30 (GSQ-30).[1]”

The MSSB comparison with the reference 1 findings were left in the discussion. “The MSSB of HLD without contracting COVID-19 or vaccination against COVID-19 was worse than previous reports of individuals with an EM rash, TBI, and PTLDS.”

In Table 2: “….long COVID…” What does it mean long COVID? How many days or months do long COVID last? For some, a week is a lot, but for others, a month is not enough.

The following information was added to clarify the uncertainty regarding the diagnostic criteria, etiology, and natural course of long COVID. “One in five HLDs who contracted COVID-19 reported long COVID.  The diagnostic criteria, etiology, and natural course of Long COVID-19 was not established at the time of the Survey.  In 2022, Subramanian and colleagues reported that 5.4% of individuals with a history of SARS and 4.3% without a history of SARS described at least one of 33 symptoms defined by the World Health Organization clinical case definition 12 week after onset of illness.”[16]

In 3.3. “One HLD who reported a severe side effect lasting more than three days improved after being administered the COVID-19 vaccine.” Was this patient revaccinated or was he still vaccinated?

The booster shots were not available at the time of the Survey. The significance of this case will need to address in future studies.

Reviewer 3 Report

Introduction must be improved. You signed incorrect data according to citation 11. According to citation 11 the frequency of side effects was 0,04%, 0,06% and 0,35% following administration of Pfizer-BioTech, Moderna and J&J/Janssen vaccines respectively. Make difference in local side effects and systematic adverse reactions including serious adverse events. You are not paying attention to medical conditions, including LD, that may result in moderate or severe immunodeficiency.

Author Response

We appreciate your thoughtful review. Your comments have strengthened the manuscript.

Introduction must be improved. You signed incorrect data according to citation 11. According to citation 11 the frequency of side effects was 0,04%, 0,06% and 0,35% following administration of Pfizer-BioTech, Moderna and J&J/Janssen vaccines respectively. Make difference in local side effects and systematic adverse reactions including serious adverse events. You are not paying attention to medical conditions, including LD, that may result in moderate or severe immunodeficiency.

The abstract cited in reference 11 only describes VAERS data.   The paper describes that incidence of side effects was 27%, 88.6%, and 48.6% for Pfizer-BioNTech, Moderna, and Johnson & Johnson's Janssen in clinical trial data.  The paragraph addressed the differences in findings from VAERS and clinical trial data: “Aside from the effects of COVID-19, its vaccine can also lead to numerous side effects. The frequency of side effects was 27%, 88.6%, and 48.6% for Pfizer-BioNTech, Moderna, and Johnson & Johnson's Janssen, respectively using clinical trial data.[11] There were 3 deaths were reported in Moderna and Janssen, but none in the Pfizer group.  None of the deaths were attributed to the vaccine. The most common side effects in the clinical trial data were injection site pain, headache, and fever, with most side effects lasting fewer than seven days.[11] In contrast, the frequency of side effects following administration of Pfizer-BioNTech, Moderna, and Johnson & Johnson’s Janssen vaccines, was of 0.04%, 0.06%, and 0.35%, respectively using a Vaccine Ad-verse Event Reporting System (VAERS) database[11].  The authors granted that the incidence of adverse events in the VAERS database may be lower due to underreporting of minor and self-limiting side effects.[11] Similar to the symptoms of COVID-19, it is not known how these side effects might compound with the burden of LD.”

The Survey addressed Fibromyalgia, Chronic fatigue, PTLDS, Mood disorder, Attention-deficit/hyperactivity disorder (ADHD), Lupus, Rheumatoid arthritis, Multiple sclerosis, and Mastocytosis.  The following section was added to address the need for research in this area. “Lyme disease has not been considered a morbidity associated with long COVID.  Comorbidities at baseline have included COPD, benign prostatic hyperplasia, fibromyalgia, anxiety, erectile dysfunction, depression, migraine, multiple sclerosis, celiac disease, and learning disability.[16] The Survey suggests the need to examine HLD as a determinate of long COVID. In particular, whether any specific metabolic or immune factors may predispose HLD to an increase in prevalence of long COVID is of particular interest.”

Round 2

Reviewer 2 Report

Most comments have been answered appropriately. Unfortunately the changes made to address my major comments are still not satisfactory although going in the right direction. Unfortunately, I did not receive a satisfactory answer to some questions.

I believe that the addition of at least one more non-HLD COVID-19 population is necessary to improve the article. But the authors say that this will only be further research.

In the discussion, the authors write that “One in five HLDs who contracted COVID-19 reported long COVID.” and in the next sentence they say that no one knows what long COVID-10 is.

I still do not understand what the authors mean by a long COVID-19? The authors write in their results that “…The vast majority of indiviuduals with a HLD were ill for more than 4 weeks (98.3%)…” and at the same time in discussion “…12 week after onset of illness”. I would like the authors to clearly write in the results or materials and methods what exactly they meant by long COVID-19 and its duration in this study.

Why am I so persistently trying to understand what is meant by a long COVID-19? This is important because Based on the duration, the authors conclude that “..the MSSB of HLD who suffered from long COVID was also significantly worse than for those who did not.”

Table 1. It is necessary to correct the HLD column for, for example, without COVID and vaccination, because all of your participants have a history of Lyme disease. Now this column raises questions.

Author Response

Thank you for your thoughtful and detailed response. We feel that your constructive criticism has resulted in an improved paper overall. We would like to respond to your points individually:

“I believe that the addition of at least one more non-HLD COVID-19 population is necessary to improve the article. But the authors say that this will only be further research.”

Regarding the addition of a non-HLD sample, we have acknowledged in the final paragraph of the discussion that an additional non-HLD group may help to understand the consequences of contracting COVID-19 or taking the COVID-19 vaccine in a population with an underlying chronic illness relative to the general population. However, we do not believe that the addition of this group to the present study is necessary to assert the present study’s aims.

This study was intended to examine a specific population – those with HLD – and to determine whether their existing symptom burden would be altered by either having contracted COVID-19 or having been vaccinated against COVID-19. In this sense, our HLD group acts as a control group. While the inclusion of a non-HLD group may help to highlight the burden of Lyme disease as a whole relative to the general population, that was not the aim of this study, and such studies can be found elsewhere. Specifically, the instrument that we selected to address the symptom load (GSQ-30) was designed and validated within an HLD population. This instrument was selected with the intention of comparing HLD populations only. It is not presently clear that inclusion of a non-HLD group would be appropriate, or that it would help to address the study’s goals. We strongly feel that this study’s goal to compare between different HLD groups has been met.

“In the discussion, the authors write that “One in five HLDs who contracted COVID-19 reported long COVID.” and in the next sentence they say that no one knows what long COVID-10 is.”

Thank you for raising this concern. We have attempted to address your point in two ways. First, we have clarified in the methods section (under 2.2. Surveys) the phrasing of the question regarding long COVID as it was asked to participants. Second, in the discussion (page 6), we have added a paragraph outlining the present state of ambiguity regarding the definition of long COVID, as there is currently no consensus on the time frame or specific symptoms. However, as noted in this addition, the World Health Organization defines 33 ongoing symptoms of COVID-19 at 12 weeks following onset of illness. We feel that this represents the best current definition of long COVID.

“Table 1. It is necessary to correct the HLD column for, for example, without COVID and vaccination, because all of your participants have a history of Lyme disease. Now this column raises questions.”

Thank you for bringing this oversight to our attention, we have corrected the table headings as requested. We hope that these additions and clarifications have satisfactorily addressed your concerns.